# π-FBG Fiber Optic Acoustic Emission Sensor for the Crack Detection of Wind Turbine Blades

**DOI:** 10.3390/s23187821

**Published:** 2023-09-12

**Authors:** Qi Yan, Xingchen Che, Shen Li, Gensheng Wang, Xiaoying Liu

**Affiliations:** 1School of Optical and Electronic Information, Huazhong University of Science and Technology, Wuhan 430074, China; 19537510017@163.com (Q.Y.); cxc19990518@163.com (X.C.); 18986252582@163.com (S.L.); 2SPIC Jiangxi Electric Power Co., Ltd., Nanchang 330096, China; wanggensheng@spic.com; 3Technology Research Institute, Shenzhen Huazhong University of Science, Shenzhen 518057, China

**Keywords:** wind turbine blades, damage detection, acoustic emission, π-FBG, WPD combined with EMD

## Abstract

Wind power is growing rapidly as a green and clean energy source. As the core part of a wind turbine, the blades are subjected to enormous stress in harsh environments over a long period of time and are therefore extremely susceptible to damage, while at the same time, they are costly, so it is important to monitor their damage in a timely manner. This paper is based on the detection of blade damage using acoustic emission signals, which can detect early minor damage and internal damage to the blades. Instead of conventional piezoelectric sensors, we use fiber optic gratings as sensing units, which have the advantage of small size and corrosion resistance. Furthermore, the sensitivity of the system is doubled by replacing the conventional FBG (fiber Bragg grating) with a π-phase-shifted FBG. For the noise problem existing in the system, this paper combines the traditional WPD (wavelet packet decomposition) denoising method with EMD (empirical mode decomposition) to achieve a better noise reduction effect. Finally, small wind turbine blades are used in the experiment and their acoustic emission signals with different damage are collected for feature analysis, which sets the stage for the subsequent detection of different damage degrees and types.

## 1. Introduction

With the scarcity of conventional fossil fuel energy sources and the growing environmental pollution problems, clean energy has gradually become a hot topic for exploration and research. Wind energy, as a kind of renewable clean energy, has received widespread attention, and the installed capacity of wind turbines has been rising sharply year by year. The new installed capacity exceeded 90 GW in 2020, which is more than a 50% increase compared with 2019 [1]. As wind turbines generally work in harsh environments for long periods of time, many damages occur, and the maintenance and repair of these damages increase their operating costs and reduce their life cycle [2]. Monitoring and timely maintenance of damage can improve the efficiency of power generation and reduce the operating cost of wind turbines [3]. As an essential component of wind turbines that are subjected to wind load for a long time, the damage rate of blades is the highest among all components and the cost accounts for about 15–20% of the total cost of wind turbines [4,5]. Damage to the blades leads to huge economic losses due to maintaining the blades, long-term downtime, reduction in power generation efficiency, and even safety accidents [6], so the damage detection of wind turbine blades is of great significance.

In recent years, a large number of detection techniques have been proposed in the field of wind turbine blade damage detection, mainly strain-based, vibration-based, image-based, and acoustic-emission-based. Strain-based detection techniques detect the slight deformation of wind turbine blades using strain sensors [7]. Lee et al. proposed a strain-sensor-based monitoring system for wind turbine blade deflection detection and proposed a new algorithm to calculate the correlation between deflection and strain [8]. The vibration-based detection method determines whether damage occurs by detecting indicators such as the intrinsic frequency and modal damping of blade vibration [9]. Dervilis et al. detected blade damage on a 9 m CX-100 blade using a pattern recognizer to identify vibrational response data. The image-based detection method analyzes captured images of wind turbine blades to determine the presence of damage through a machine vision approach [10]. Stokkeland et al. proposed an autonomous machine vision method for identifying and tracking wind turbines as well as blades using an UAV (unmanned aerial vehicle), which uses the Hough line transform as the recognition algorithm and the Kalman filter for tracking [11]. Acoustic-emission-based detection methods detect damage by detecting transient elastic waves released due to internal stress redistribution during blade damage [12]. Bouzid et al. integrated the techniques of acoustic emission and wireless technology to propose a structural health monitoring system for monitoring wind turbine blades in the field [13].

Compared to other techniques, acoustic emission detection technology is more sensitive and can provide a comprehensive detection of the complete structure as well as the internal damage of the structure, especially to detect the internal early subtle damage and the extension process of the damage [14]. The analysis of the characteristics of acoustic emission signals such as amplitude, energy, rise time, and root mean square value can further detect the state of the wind turbine blade, the degree of damage, and the propagation process [15]. Therefore, acoustic-emission-based wind turbine blade detection is of great research value and an extremely promising technique, and is investigated in this paper.

Conventional acoustic emission detection techniques use piezoelectric sensors, but they have the disadvantage of being large in size and mass and susceptible to electromagnetic interference. Some researchers are now starting to use fiber gratings as acoustic emission sensors, which have advantages such as small size, light quality, corrosion resistance, resistance to electromagnetic interference, and high sensitivity. Therefore, fiber optic acoustic emission detection technology is considered as a potential and emerging technology in the field of structural health detection [16]. In the early twenty-first century, Perez et al. verified that simulated acoustic emission events could be detected using fiber grating sensors [17]. Vidakovic M et al. used FBG instead of PZT sensors that cannot be applied to seawater environments to conduct acoustic emission testing of metal materials in marine applications [18].

In this paper, we investigate the application of fiber grating acoustic emission detection techniques to wind turbine blade damage detection. First, a fiber grating acoustic emission detection system is built, and the sensitivity of the system is further improved by replacing the FBG with π-FBG. For the noise in the signal, WPD denoising is combined with EMD to achieve better denoising results. Finally, different levels of damage are simulated on small wind turbine blades, and their energy characteristics are analyzed.

## 2. FBG Acoustic Emission Detection System

### 2.1. Fundamentals of FBG Acoustic Emission Detection

The FBG is a diffraction grating formed by the periodic distribution of the refractive index of the fiber core and fabricated normally using the phase mask method [19]. Its effect is equivalent to a narrowband filter or reflector, as shown in Figure 1.

The central wavelength of the FBG reflection spectrum is given by
(1)$$\lambda_{B}=2n_{eff}\Lambda$$
where $n_{eff}$ is the effective refractive index of the core and Λ is the grating period of the FBG. When a fiber grating is subjected to an external signal, it undergoes micro-deformation, so its grating period changes, which eventually leads to a drift in its central wavelength. Therefore, we can detect the corresponding signal by measuring its central wavelength or converting it to other signal changes.

### 2.2. FBG Acoustic Emission Detection System

A tunable narrowband laser is used to convert the drift of the FBG center wavelength into the change in light intensity, which is then detected using a photodetector. The schematic diagram and principle of the system are shown in Figure 2.

As shown in Figure 2, the narrowband laser is reflected by the fiber grating and then converted into an electrical signal by a photodetector, which is amplified and fed into the oscilloscope. From the spectrogram, the area of the overlap between the narrowband laser spectrum and the FBG reflection spectrum represents the light intensity detected by the photodetector. Under the action of an acoustic emission signal, the center wavelength of the FBG drifts, resulting in a change in its overlapping area, which is finally transformed into the signal of light intensity change detected by the photodetector [20]. The final voltage signal $V_{S}$ detected by the photodetector can be expressed as
(2)$$V_{s}=\alpha IK_{s}\Delta \lambda_{B}$$
where α is the proportionality constant, *I* is the laser power, $K_{s}$ is the slope of the grating reflection spectrum, and $\Delta \lambda_{B}$ is the grating center wavelength drift. It can be seen that the higher the slope of the reflection spectrum, the higher the sensitivity of the system. Therefore, this paper uses a π-FBG to replace the FBG. A π -FBG introduces a phase mutation point of π in the grating region, which can be regarded as a cascade of two FBGs, resulting in an extremely narrow transmission peak in the center of its reflection spectrum, as shown in Figure 3b. The narrowband transmission peak in the middle of it has a higher slope, so the narrowband laser is aimed at the midpoint of the sloping edge of the transmission peak to obtain a higher system sensitivity in this paper.

### 2.3. Experimental Device

The light source used in the experiment is a tunable narrowband laser, model TLG-210. The output band is 1528~1567 nm, the output power range is +7~+15.5 dBm, and the output channels 1–4 are optional.

The photodetector used in this system is an ET-3000A from EOT Company in the United States. This photodetector has a built-in preamplifier function, an amplification factor of 26 dB, a saturated optical power of −20 dBm, and a bandwidth of 30 KHz~1.5 GHz.

The oscilloscope model used in the experiment is a DPO2012. The oscilloscope has a simulated bandwidth of 100 MHz, two analog channels, a maximum sampling rate of 1 GS/s, a maximum ability to collect 1 M point data, and a maximum waveform capture rate of 5000 wfm/s.

### 2.4. System Testing

In order to confirm the sensitivity of the π-FBG, this paper used tape to attach the FBG and π-FBG to the steel plate with a coupling agent applied in between to reduce the signal loss, and then built the sensing system separately. The center wavelength of the experimental FBG is 1549.970 nm with a 3 dB bandwidth of 0.23 nm, and the center wavelength of the π-FBG is 1549.985 nm with a 3 dB bandwidth of 0.3 nm. We adjusted the center wavelength of the tunable narrowband laser to align with the maximum slope of the reflection spectrum of both, and kept the other conditions the same. The PZT was used to generate a standard sinusoidal signal at 100 KHz, and both FBGs were located at a distance of 10 cm from the PZT. The final signals acquired by both systems are shown in Figure 4, and it can be seen that both can detect the signals, but the π-FBG is about twice as sensitive as the FBG and has less distortion.

Further, we broke a pencil lead to simulate the acoustic emission signal to test whether the system can detect tiny acoustic emission signals. According to the requirements of the national standard GB/T18182-2000, the lead breaking signal (breaking a pencil lead) is used to simulate the acoustic emission signal. An HB 0.5 mm automatic pencil was used, and the pencil lead of the automatic pencil was extended by 2.5 mm, breaking at an angle of 30 with the steel plate surface. The pencil breaking position was 10 cm away from the fiber optic grating sensor. We also purchased a standard piezoelectric acoustic emission sensor for simultaneous detection for comparison. The pencil lead breaking point was 10 cm from the two sensors, respectively. The relative placement relationship of the experimental instruments is shown in Figure 5. From left to right, these are the piezoelectric sensor, the automatic pencil, and the fiber optic grating sensor.

The detected signals and their frequency spectrum are shown in Figure 6. It can be seen from the figure that the detected signals of the two are relatively consistent, and the larger component of about 40 KHz in the measurement signal of the acoustic emission sensor is caused by its own resonance frequency. It can be demonstrated that this system can detect the acoustic emission signal, but the signal contains large noise that needs to be processed.

## 3. Signal Denoising

### 3.1. Wavelet Packet Decomposition

Wavelet analysis is a typical time–frequency domain analysis method applicable to non-smooth burst signals like acoustic emission signals, but it only decomposes the signal at low frequencies and has poor resolution at high frequencies. In this case, Wickerhauser et al. further optimized the original wavelet transform based on its theory and developed WPD, which can decompose the low-frequency part and the high-frequency part of the signal at the same time [21]. Compared with the wavelet transform, the frequency resolution of the high-frequency signal is well ensured, and it also improves the analysis of the signal by adaptively selecting the frequency band according to the signal characteristics.

From the perspective of function theory, WPD is the projection of a signal into a space composed of wavelet packet basis functions, and from the perspective of signal processing, it is the decomposition of a signal through a series of filters with different center frequencies but the same bandwidth. It decomposes the signal into high-frequency subbands and low-frequency subbands step by step, which finally forms a subband tree structure. The diagram of a third-order wavelet packet decomposition tree is shown in Figure 7, where A represents the approximate filtering, which is the low-frequency part of the higher-level signal, and D represents the detail filtering, which is the high-frequency part of the higher-level signal.

The steps for noise reduction of the signal through wavelet packet decomposition are:Select a suitable wavelet basis function and determine the number of decomposition layers of the signal. For acoustic emission signals, wavelet basis function generally adopts Daubechies wavelets, Symlet wavelets, and Coiflet wavelets. The blade acoustic emission signal is generally a high-frequency burst signal, so a wavelet with high compactness, good orthogonality, and high vanishing distance should be chosen. When the number of decomposition layers is higher, the distortion of the reconstructed signal is bigger while the computation is increased, and when the number of layers is too small, the noise cannot be eliminated well.Calculate the wavelet packet coefficients of each layer after signal decomposition, and then determine the optimal signal decomposition path by minimizing a cost function, which is also called the optimal tree. In this paper, the minimum Shannon entropy criterion is adopted, and f is the original signal and $f_{i}$ is the wavelet packet coefficient after signal decomposition. The entropy calculation formula is
(3)$$E\left[f\right]=-\sum_{i}f_{i}^{2}log\left[f_{i}^{2}\right]$$

In the process of signal wavelet packet decomposition, when the entropy of the parent node is less than the entropy of the two child nodes, the decomposition will not continue, so as to obtain the final optimal wavelet packet decomposition tree.
3.Select a suitable threshold for filtering the wavelet decomposition coefficients. The selection of threshold rules and functions determines the quality of noise reduction. Currently, commonly used threshold rules include sqtwolog rules, rigrsure rules, heursure rules, and minimaxi rules. In this paper, the minimaxi threshold rule is adopted. When the signal has only a small number of high-frequency parts in the noise range, the effect is better, and the loss of the original signal is small while denoising. It generates an extreme value with the goal of minimizing the mean square error, which can minimize the maximum mean square error. The threshold calculation expression is
(4)$$\lambda =\left\{\begin{array}{cc} \sigma \left(0.396+0.1829{log}_{2}N\right) & N\geq 32\\ 0 & N<32 \end{array}\right.$$
where *N* is the number of wavelet packet coefficients in the maximum decomposition layer of the WPD. σ is the standard deviation of the noise signal, and the expression is
(5)$$\sigma =\frac{\sum_{i=1}^{N}\left|f_{i}^{M}\right|}{0.6745N}$$
where $f_{i}$ is the *i*th layer wavelet packet decomposition coefficient. Currently, the commonly used thresholding functions are mainly hard and soft thresholding functions. The soft threshold function subtracts the threshold when the coefficient is greater than the threshold and sets it to zero when it is less than the threshold, which is smoother, but may lose some features. The hard threshold function retains the coefficients larger than the threshold and sets them to zero when they are smaller than the threshold, which can retain the signal characteristics, but the smoothness is lacking. In this paper, the soft threshold function is selected, and its expression is
(6)$$w_{\lambda }=\left\{\begin{array}{cc} \left[sgn\left(w\right)\right]\left(\left|w\right|-\lambda \right) & \left|w\right|\geq \lambda \\ 0 & \left|w\right|\geq \lambda \end{array}\right.$$
4.Reconstruct the wavelet coefficients after processing.

### 3.2. Wavelet Packet Decomposition Combined with Empirical Modal Decomposition

WPD requires a pre-selected wavelet basis function, the choice of which has a significant impact on the overall analysis results. Even though the wavelet basis may be optimal globally, it may not be optimal locally and lacks adaptivity.

EMD is a signal analysis method proposed by Huang [22]. It decomposes according to the time scale characteristics of the signal itself, without pre-setting any basis functions, it has self-adaptation, can obtain different modes of the signal, and has obvious advantages in processing non-stationary nonlinear data. The EMD decomposition decomposes the nonlinear wave signal into a set of narrow-bandwidth stationary signals with time–frequency characteristics similar to cosine and a residual term. The signal frequencies are arranged in order from high to low, which is called intrinsic mode function (IMF).

In this paper, the WPD noise reduction is combined with EMD to improve its self-adaptability, and the final noise reduction steps are:Use EMD to decompose the signal into IMF components and residuals.Calculate the correlation coefficient of each IMF component with the original signal, and select the IMF component with the larger correlation coefficient.Select the appropriate wavelet basis function to perform WPD noise reduction on the selected IMF components.Reconstruct the denoised IMF component.

### 3.3. Noise Reduction Test

In order to verify the effect of comparing the above noise reduction methods, an acoustic emission signal was simulated in this paper, and the expression is
(7)$$x\left(t\right)=\exp\left(-2\pi {\left(\frac{t-t_{1}}{\lambda_{1}}\right)}^{2}\right)\sin\left(2\pi f_{1}(t-t_{1})\right)\;+\exp\left(-2\pi {\left(\frac{t-t_{2}}{\lambda_{2}}\right)}^{2}\right)sin(2\pi f_{2}(t-t_{2})))\;+\exp\left(-2\pi {\left(\frac{t-t_{3}}{\lambda_{3}}\right)}^{2}\right)\sin\left(2\pi f_{3}(t-t_{3})\right)$$
where f_1_ = 90 KHz, *f*_2_ = 100 KHz, f_3_ = 130 KHz, t_1_ = 0.4 ms, t_2_ = 0.5 ms, t_3_ = 0.6 ms, $\lambda_{1}$ = 0.0001, $\lambda_{2}$ = 0.00015, $\lambda_{3}$ = 0.00018, the sampling rate is 1024 KHz, and the number of sampling points is 1024. The waveform and the frequency spectrum are shown in Figure 8.

The Gauss white noise of SNR (signal-to-noise ratio)-4 dB was added to the simulation signal, and the waveform and the frequency spectrum of the simulation signal are shown in Figure 9.

We used the WPD noise reduction method to denoise the signal, selected db4 wavelets for the wavelet base function, and selected five layers for the number of decomposition layers. The signal waveform and frequency spectrum after noise reduction are shown in Figure 10.

We used WPD combined with the EMD method to denoise the signal, selected IMF2 and IMF3 components for WPD denoising, respectively, selected db4 wavelets and db3 wavelets for wavelet base, respectively, and selected five layers for the number of decomposition layers. The signal waveform and spectrum after noise reduction are shown in Figure 11.

It can be seen from the figure that the wavelet packet combined with EMD denoising is more effective and less distorted. This paper quantitatively measures the noise reduction effect through *SNR* and *RMSE* (root mean square error), which are defined as follows:(8)$$SNR=-10lg\frac{\sum_{i}^{n}{\left(x\left(i\right)-\hat{x}\left(i\right)\right)}^{2}}{\sum_{i}^{n}x^{2}\left(i\right)}$$
(9)$$RMSE=\sqrt{\frac{1}{n}\sum_{i=1}^{n}{\left(x\left(i\right)-\hat{x}\left(i\right)\right)}^{2}}$$
where $x\left(i\right)$ is the original signal and $\hat{x}\left(i\right)$ is the denoised signal. The larger the *SNR*, the smaller the *RMSE*, and the better the noise reduction effect. We calculated the *SNR* and *RMSE* of the denoised signal using the two methods, respectively, and the results are shown in Table 1.

## 4. Wind Turbine Blade Damage Acoustic Emission Signal Characteristics

### 4.1. Wind Turbine Blade Structure

This paper takes a small wind turbine blade as an example to study under laboratory conditions, as shown in Figure 12. The wind turbine blade is mainly composed of a shell and an internal support. The support layer is made of glass fiber composite material, and the shell is wrapped with a resin protective layer and surface coating on the outside of the glass fiber composite material layer.

### 4.2. Wind Turbine Blade Damage Acoustic Signal

To collect the acoustic emission signal from the damage of the glass fiber composite, the main material of the wind turbine blade, we cut the wind turbine blade and used tweezers to break the internal support layer to generate the acoustic emission signal, as shown in Figure 13. We stuck the π-FBG onto the surface of the blade with tape, 10 cm away from the crack, applied coupling agent, and connected the experimental instrument according to the experimental system design diagram. The narrowband light source wavelength was adjusted to calibrate the sensitivity of the FBG acoustic emission sensor.

The signal was denoised using WPD combined with the EMD method, and the original signal and the denoised signal are shown in Figure 14. It can be seen that the denoising algorithm used in this paper can effectively remove the noise in the signal and the low-frequency vibration signal introduced during the operation. From the spectrum, it can be seen that the acoustic emission signal of the glass fiber composite damage is mainly concentrated in the range of 70–140 KHz, with the peak frequency around 95 KHz.

Further, we investigated the acoustic emission signal during the overall damage of the turbine blade. We cut a crack in the turbine blade, fixed it at one end, and then applied pressure to the other end to expand the crack, as shown in Figure 15. We applied a π-phase-shifted FBG to the blade surface at a distance of 10 cm from the crack and used the system to collect the acoustic emission signal during this process.

We divide the whole process into three stages: front, middle, and back. The signal after denoising in the three stages is shown in Figure 16. It can be seen from the figure that as the stage progresses, the high-frequency signal in the signal gradually increases, and the signal distribution in the later stage is similar to the above-mentioned glass fiber damage signal. This is because the initial damage is mainly the resin protective layer of the blade, and its frequency is low. As the damage spreads, the glass fiber layer of the housing gradually begins to be damaged, and the signal gradually begins to contain high-frequency signals generated by glass fiber damage. When the damage is serious in the later stage, it is mainly the damage signal of the glass fiber of the shell and the support layer.

The experimental results show that the main frequency of the signal during initial surface cracking is mainly concentrated between 0 KHz and 70 KHz, with a peak frequency between 35 KHz and 40 KHz. The signal frequency of the internal fiber layer fracture is mainly concentrated between 70 KHz and 140 KHz, with a peak frequency between 95 KHz and 100 KHz. It can be seen that there are certain differences in the frequency range of these two different types of damage. Since WPD can be seen as decomposing the signal into different frequency bands, we calculated the signal energy distribution in each band after decomposing the denoised signal. We used db5 wavelets to decompose the signal into four layers, which are divided into 16 frequency bands. The system sampling rate is 1250 KHz, so the bandwidth of each band is about 39 KHz. Since the signal component is almost zero in the last 8 bands, the first 8 bands are taken, as shown in Figure 17. It is more obvious from the figure that the high-frequency energy in the signal gradually increases as the damage process progresses.

## 5. Conclusions

In this paper, we construct an acoustic emission signal sensing system using a fiber grating to study wind turbine blade damage signals. To enhance sensitivity, a π-phase-shifted fiber Bragg grating (π-FBG) is adopted in place of the conventional FBG, a decision made following experimental comparisons. It is important to highlight that the temperature’s influence on FBG sensors is acknowledged; thus, all experimentation and data collection mentioned in this article occurred at a controlled temperature of 25 degrees Celsius.

Addressing the issue of system noise, an enhanced mode decomposition (EMD) coupled with wavelet packet decomposition (WPD) denoising technique is employed, leading to more effective noise reduction. This study focuses on small wind turbine blade assessment. Through experimental acquisition and analysis of acoustic emission signals resulting from blade damage, distinctions in the energy distribution of these signals based on blade materials and degrees of damage are observed. This serves as a foundational insight for future fiber grating acoustic emission detection, facilitating classification of diverse types of blade damage.

The system’s potential for industrial application can be further refined. Furthermore, extensive experimentation on larger wind turbine blades during production can yield various acoustic emission signals linked to distinct types of damage. Analyzing and researching these signals contributes to the evolution of sophisticated detection.

## Figures and Tables

**Figure 1 sensors-23-07821-f001:**
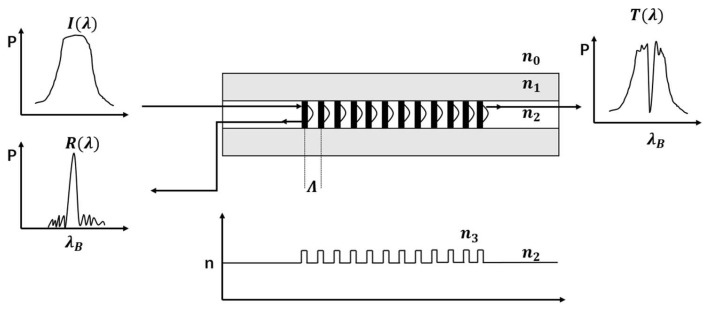
Refractive index profile and spectral response of an FBG.

**Figure 2 sensors-23-07821-f002:**
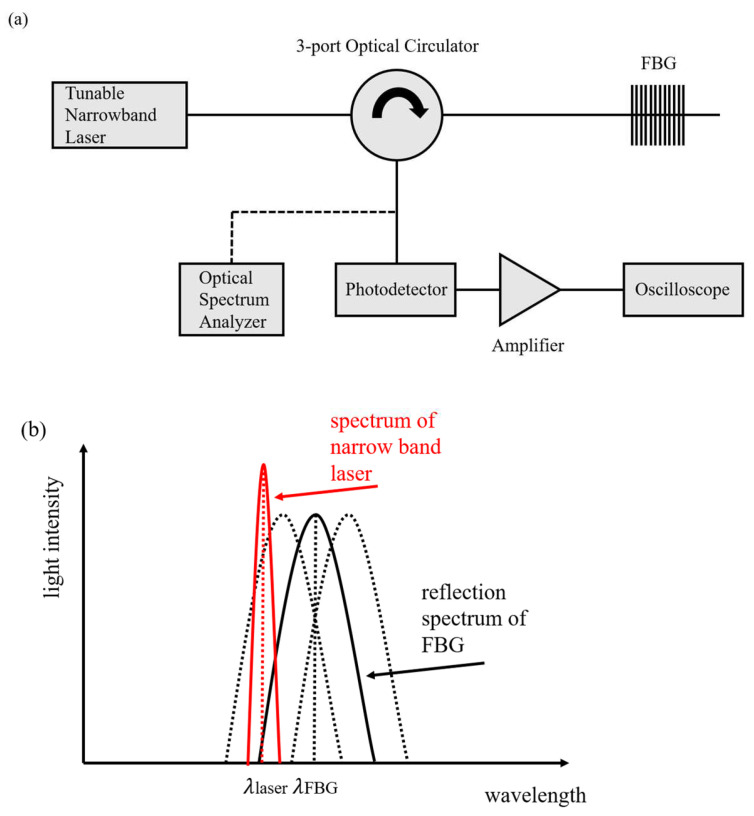
(**a**) Fiber grating acoustic emission detection system. (**b**) Schematic diagram of the principle.

**Figure 3 sensors-23-07821-f003:**
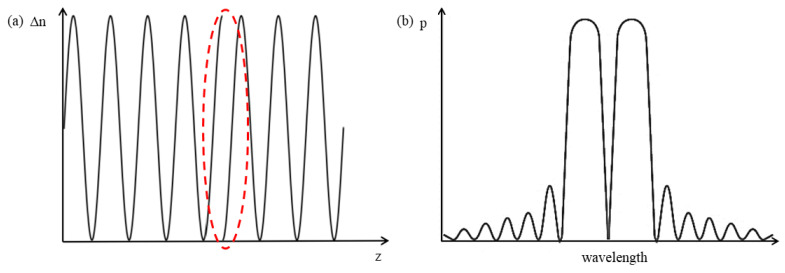
(**a**) Refractive index distribution of π-FBG. (**b**) Reflection spectrum of π-FBG.

**Figure 4 sensors-23-07821-f004:**
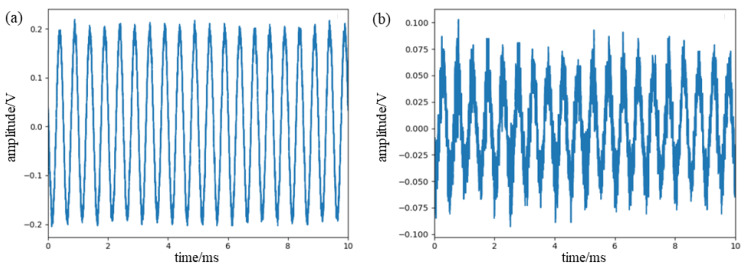
(**a**) Signal detected by π-FBG. (**b**) Signal detected by FBG.

**Figure 5 sensors-23-07821-f005:**
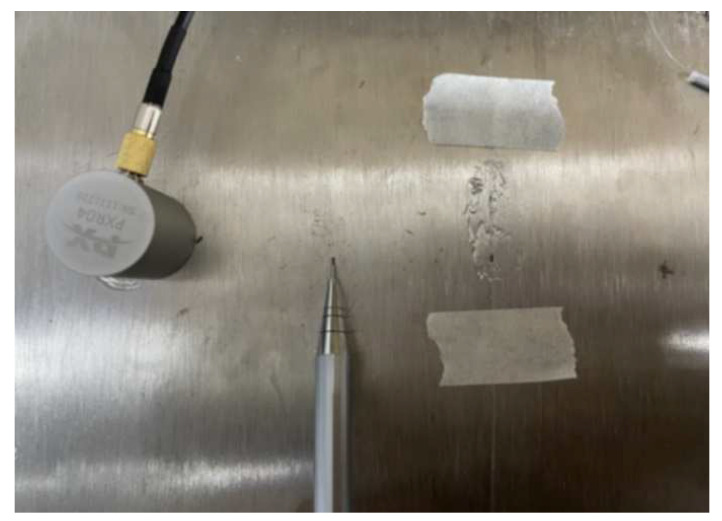
Schematic diagram of simulated acoustic emission signal from broken lead.

**Figure 6 sensors-23-07821-f006:**
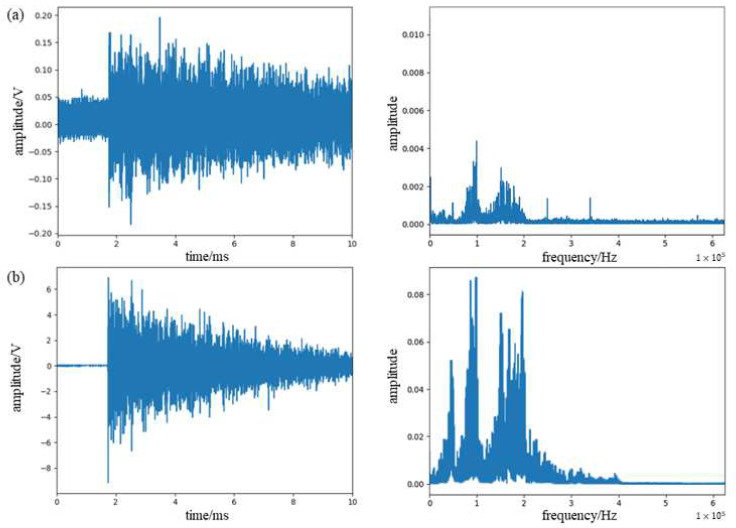
(**a**) Waveform spectra and spectrograms of lead break signals detected by π-FBG. (**b**) Waveform spectra and spectrograms of lead break signals detected by PZT.

**Figure 7 sensors-23-07821-f007:**
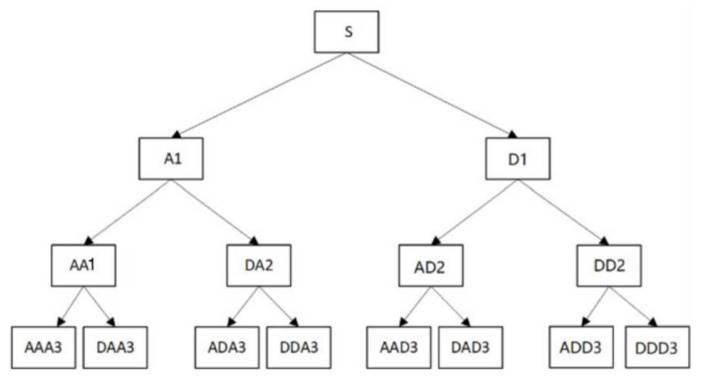
Schematic diagram of three-layer wavelet packet decomposition.

**Figure 8 sensors-23-07821-f008:**
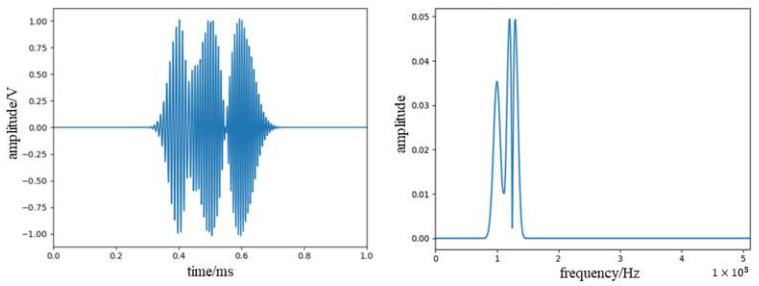
Waveform spectra and spectrograms of simulated acoustic emission signal.

**Figure 9 sensors-23-07821-f009:**
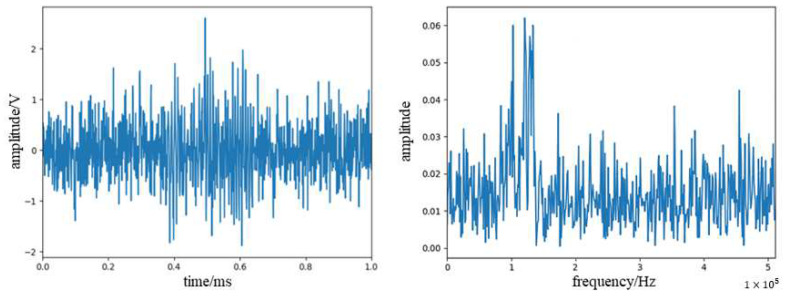
Waveform spectra and spectrograms of simulated acoustic emission signal after adding noise.

**Figure 10 sensors-23-07821-f010:**
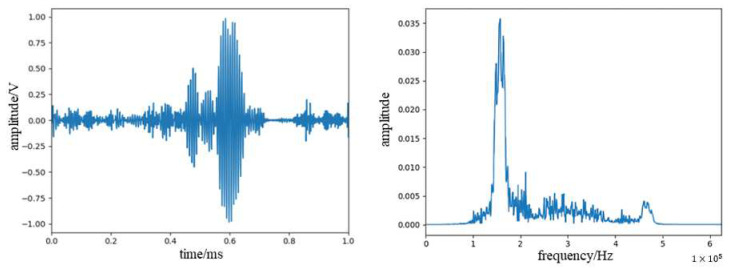
Waveform spectra and spectrograms of acoustic emission signal after WPD noise reduction.

**Figure 11 sensors-23-07821-f011:**
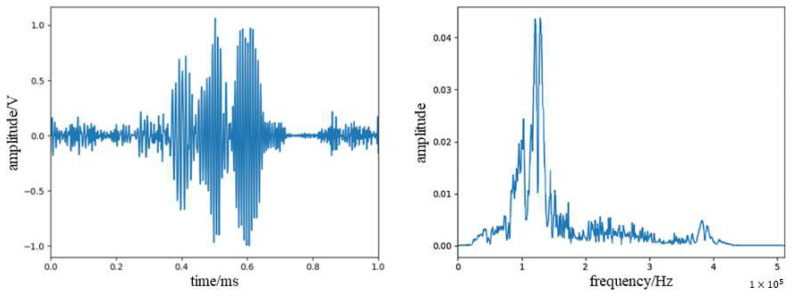
Waveform spectra and spectrograms of acoustic emission signal after WPD combined with EMD noise reduction.

**Figure 12 sensors-23-07821-f012:**
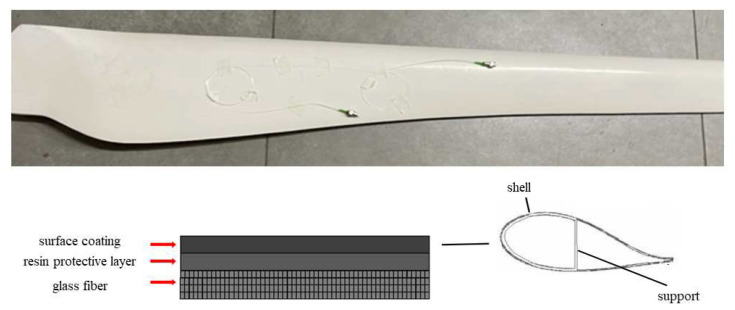
Small wind turbine blade.

**Figure 13 sensors-23-07821-f013:**
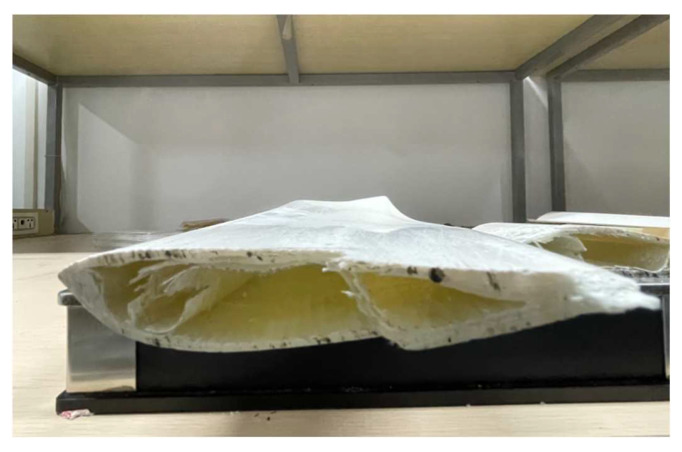
Wind turbine blade support layer damage.

**Figure 14 sensors-23-07821-f014:**
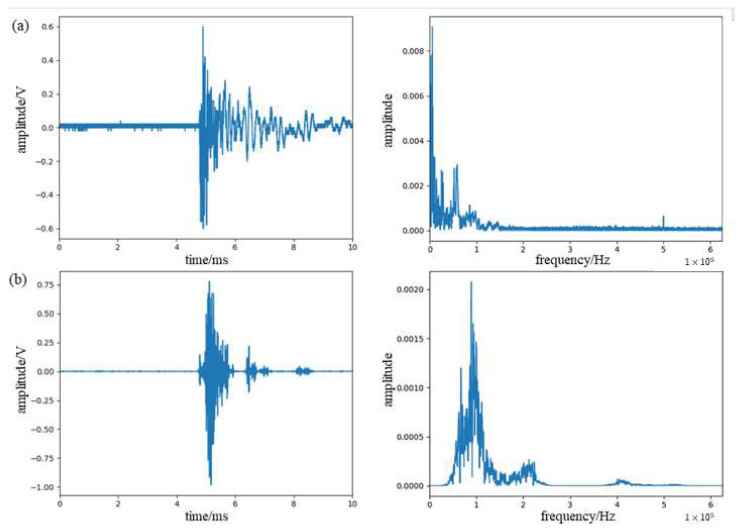
(**a**) Waveform spectra and spectrograms of acoustic emission signals of glass fiber composite damage. (**b**) Waveform spectra and spectrograms of acoustic emission signals of glass fiber composite damage after noise reduction.

**Figure 15 sensors-23-07821-f015:**
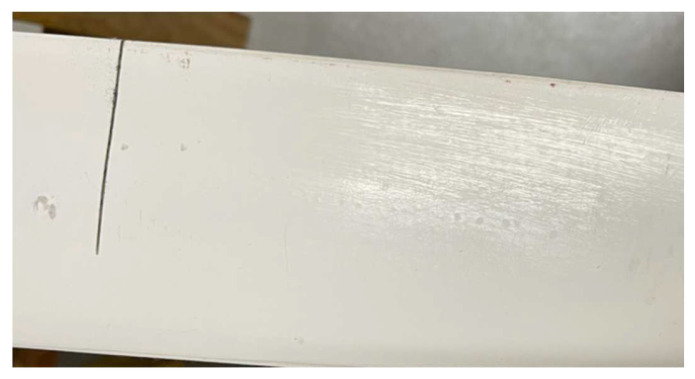
Schematic diagram of wind turbine blade damage simulation.

**Figure 16 sensors-23-07821-f016:**
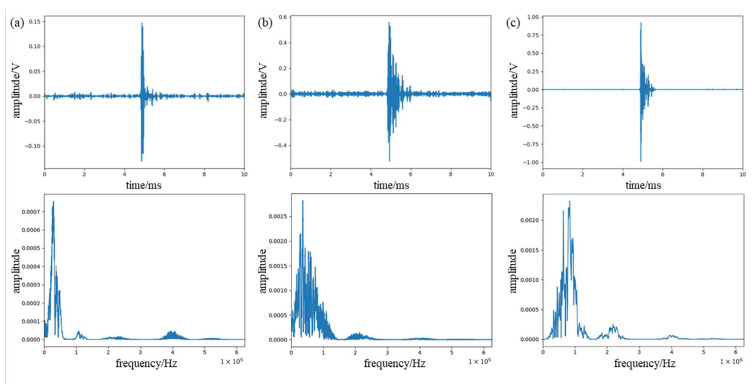
(**a**) Acoustic emission signal of early damage of the blade. (**b**) Acoustic emission signal of mid-stage damage of the blade. (**c**) Acoustic emission signal of late damage of the blade.

**Figure 17 sensors-23-07821-f017:**
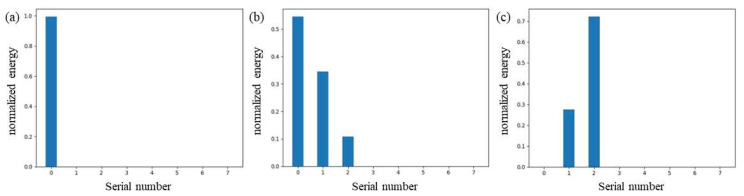
(**a**) Acoustic emission signal energy characteristic distribution of early damage of the blade. (**b**) Acoustic emission signal energy characteristic distribution of mid-stage damage of the blade. (**c**) Acoustic emission signal energy characteristic distribution of late damage of the blade.

**Table 1 sensors-23-07821-t001:** Indicators of the denoised signal.

Method	*SNR* (dB)	*RMSE*
WPD	2.261	7.043
WPD + EMD	0.228	0.129

## Data Availability

Data is available upon request by contacting correspondence.

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
