# Peer review of "π-FBG Fiber Optic Acoustic Emission Sensor for the Crack Detection of Wind Turbine Blades"

_sensors, 2023, doi:10.3390/s23187821_

Round 1
Author Response
Dear Reviewer,
First of all, I would like to express my gratitude for your valuable comments on this article, which have enabled me and our team to realize the shortcomings in the content. Secondly, in response to the issues you raised, I have made specific revisions.
Firstly, I have added a detailed introduction to the experimental instruments in the second chapter of the article, including the parameters of the laser, photoelectric detector, and oscilloscope we used. Now you can directly obtain this information from the article.
Secondly, regarding the issue of signal attenuation, when a fault occurs, the acoustic emission signal propagates along the surface of the blade, which is inherently a continuously attenuating wave. As for the dynamic range you mentioned, it is related to the 3dB bandwidth of the fiber Bragg grating sensor itself and the scanning rate of the spectrometer. Moreover, larger detection ranges can be achieved by using an array of fiber Bragg grating sensors with different center wavelengths.
Thirdly, concerning the temperature influence factors you mentioned, this is indeed very important. We have kept the laboratory temperature fixed at 25 degrees Celsius from the beginning of the experiment until the end. Temperature does have an impact on the fiber Bragg grating sensor, and we have made corresponding temperature compensation. However, due to limited space in the article, it was not fully displayed and thus overlooked. I apologize for this and have indicated the experimental temperature in the summary section of the article. Of course, if it is an actual application product, there will definitely be a corresponding temperature compensation grating to compensate and correct the detection results.
The fourth point is regarding the noise issue you mentioned. During the actual operation, factors such as blade deformation and low-frequency vibrations may affect the detection results. When applied to wind turbine blades in practice, the on-site noise situation can be very complex and cannot be described using a single mathematical model. Therefore, in order to address the noise problem in the system, we combines the traditional wavelet packet decomposition (WPD) denoising method with empirical mode decomposition (EMD) to achieve better denoising effect. Furthermore, in future product development, we will consider more types of external noise signals.
Once again, thank you for your patience and feedback. If there are any other areas that need improvement, please let me know, and I will strive to enhance the article further.
Sincerely,
Xingchen Che
Reviewer 2 Report
The authors present the use of a π-shifted FBG attached to a wind turbine blade interrogated in such a way that Acoustic Emission Detection can take place. The signal is denoised using Wavelet Packet Decomposition along with another method mentioned as Empirical Mode Decomposition.
The authors first test the performance of the denoising technique on a steel plate, by generating a noise signal and by performing the pencil lead break test using both a standard and π-shifted FBG.
The authors then use a π-shifted FBG for crack detection on a wind turbine blade. Although the title of this work is centered around this test, this section is actually the most weakly presented and lacks key information that is required to be provided before the paper is recommended for publication.
Initially, the authors must provide infromation of the hardware used to interrogate the π-shifted FBG. The reader needs to know the spectral linewidth of the tunable laser used and the photodiode speed and responsiveness in order to understand the acquired results.
Furthermore, there should be a detailed discussion on the way the optical fiber was attached to the wind turbine blade. In the text the orientation of the optical fiber and the attachment means (glue? embossing? scotched?) is missing and it is crucial to the reader, as is can act as an acoustic filter (there is always a transfer function between the acoustic waves and the optical fiber). Therefore, it is extremely important for the reader to know the exact orientation of the optical fiber with respect to the acoustic wave source and the attachment means.
In addition, the authors need to comment on the acoustic frequency response of the optical fiber. Is there no audible signal detection? If not why? Breaking the wind turbine blade typically causes an audible noise.
The authors should also discuss in more detail the choice of parameters in the WPD decomposition and the rationale behind them. If the signal included an audible part, would the selection be the same?
Finally, the authors should rephrase the conculsions section as the use of English in that section prevents the reader from understanding the message the authors want to convey.
The document, and especially the conclusions section, would greatly benefit from English language editing, as the aforementioned section is hard to understand.
Author Response
Dear Reviewer,
First and foremost, I would like to express my sincere gratitude for the invaluable feedback you have provided on this paper. Your insights have been instrumental in helping both myself and my team recognize the inadequacies within the manuscript. In light of the issues you raise, I have undertaken specific modifications.
To address your concerns, I have incorporated a comprehensive description of the experimental apparatus in the second chapter of the article. This description encompasses the parameters of both the laser and the photodetector we employ, allowing the reader direct access to this critical information within the paper.
Regarding the connection method between the fiber grating sensor and the wind turbine blade, we adopt an adhesive-based approach. In addition, we apply a coupling agent between the two to minimize signal loss. I have elaborated on these details in Section 2.4.
Also, about the transfer function and the frequency of the sound you raised.. We employ fiber grating sensors to detect acoustic emission signals by capitalizing on their sensitivity to stress variations. A minute change in stress induces a change in the grating period of the fiber, which affects its central wavelength and facilitates detection. In practical scenarios, when faults such as blade cracks occur in a wind turbine blade, the acoustic signal propagates as a decaying oscillation along the blade surface. This phenomenon triggers fluctuations in the central wavelength of the fiber optic grating sensor. By employing an array of these sensors, we are able to perceive varying magnitudes of oscillations, allowing us to pinpoint the source of the acoustic emission. Subsequent experiments showed that the main frequency range of the signal when the wind turbine surface cracks is between 0 kHz and 70 kHz, with a peak frequency between 35 kHz and 40 kHz. On the other hand, signals related to internal fiber layer fracture mainly span between 70kHz and 140kHz, with peak frequencies ranging from 95kHz to 100kHz. Importantly, these frequency ranges encompass the range of human hearing, and I have incorporated this information into the paper.
I wish to reiterate my appreciation for your patience and thoughtful suggestions. Should there remain any further areas that require refinement, I kindly request that you bring them to my attention. I am fully committed to enhancing this article to its fullest potential.
Thank you again for your guidance.
Warm regards,
Xingchen Che
Reviewer 3 Report
Comment to Authors.
1. It would have been good to have a description within the paper of the specifications of the narrow linewidth laser used in the study and demonstration of the pi-FBG AE detection.
2. It would be good to address the temperature and strain sensitivity of the pi-FBG and its effect on the narrow linewidth laser AE detection system.
3. It would be good to address the solution for monitoring the pi-FBG AE detection sensitivity in the presence of high strain and temperature variations commonly found in windmill blade structures.
4. Comment in the Figure 5 description it is mentioned that the distance from the pencil lead break and the FBG sensors is 10-cm as shown in the Figure, however, within the Figure the location of the pencil lead break event and the FBG sensor does not look like 10-cm. It looks more like 2-cm.
5. In the paper line 221 there is a change in the font.
Author Response
Dear Reviewer,
Firstly, I would like to express my gratitude for the valuable feedback on this paper, which helped me and my team identify areas for improvement. Secondly, I have addressed the issues you raised by making specific revisions.
First, I have included a detailed description of the experimental setup, including the parameters of the laser used, in the second chapter of the article. This will allow you to access this information directly within the article.
Second, for the problem that the distance between the sensor and the automatic pencil in the picture does not seem to meet the description of 10cm, the wide field of view requirement (20cm range) for the two different sensors during testing made it challenging to accurately capture the images within the required range. In order to facilitate clear imaging (clearly display the piezoelectric sensor model used) and more easily crop the image, we have decided to place the three closer together and capture them in the same frame as a schematic. This image is only shown as a schematic diagram of the relative positions between the piezoelectric sensor, the lead breaking point, and the fiber optic grating sensor. It is important to clarify that the actual measurements were strictly conducted following the 10cm standard specified in the national standard GB/T18182-2000. I apologize for not providing this explanation in detail in the article, which led to misunderstandings. I have promptly made the necessary corrections.
Thirdly, the article's font has been standardized.
Finally, I would like to extend my appreciation for your patience and valuable suggestions. If there are any further areas for improvement, do not hesitate to point them out. I will strive to further enhance the quality of the article.
Thank you once again.
Sincerely,
Xingchen Che
Round 2
Reviewer 1 Report
The authors have improved the paper. The description of the experiment was added. But, I would recommend the authors to exclude photos of the equipment (fig.4, 5 and 6) from the paper.
Author Response
Dear Reviewer,
I would like to express my sincere gratitude for taking the time to provide feedback and suggestions on my article. Your insights have been invaluable in refining the content and ensuring its readiness for publication. Moreover, I have deleted the experimental equipment diagram from the article.
Once again, thank you for your thoughtful guidance and support. Your input has been instrumental in shaping the quality of the article, and I truly appreciate your contribution.
Sincerely,
Xingchen Che
Reviewer 2 Report
The authors have addressed most of the original review requests to improve the paper quality and this is evident in throughout the paper.
However, I would still recommend English proofreading, as some parts of this work, especially the conclusions, require some rewording to become more clear.
The paper can be recommended for publication after this minor revision.
Please, see previous comment on English proofreading. It is still required in some parts of the paper.
Author Response
Dear Reviewer,
I would like to express my sincere gratitude for taking the time to provide feedback and suggestions on my article. Your insights have been invaluable in refining the content and ensuring its readiness for publication. I am committed to further enhancing the article, particularly in terms of English grammar, to ensure that the details are both comprehensive and accurate.
Once again, thank you for your thoughtful guidance and support. Your input has been instrumental in shaping the quality of the article, and I truly appreciate your contribution.
Sincerely,
Xingchen Che